# Interleukin-6-Production Is Responsible for Induction of Hepatic Synthesis of Several Chemokines as Acute-Phase Mediators in Two Animal Models: Possible Significance for Interpretation of Laboratory Changes in Severely Ill Patients

**DOI:** 10.3390/biology11030470

**Published:** 2022-03-18

**Authors:** Ihtzaz Ahmed Malik, Giuliano Ramadori

**Affiliations:** 1Department of Geriatrics, University Medical Center Goettingen, Waldweg 33, D-37073 Goettingen, Germany; 2Center of Internal Medicine, University Medical Center Goettingen, Robert-Koch Str. 38, D-37075 Goettingen, Germany; giulianoramadori@gmail.com

**Keywords:** SARS-CoV-2, interleukin-6, cytokines, chemokines, liver, inflammation, acute-phase reaction, acute-phase chemokines

## Abstract

**Simple Summary:**

The release of acute-phase proteins and cytokine storms are considered critical parameters for the progression of COVID-19 disease. The increase in the serum levels of cytokines such as IL6 and IL8 observed in patients primarily infected with the SARS-CoV-2 virus has been used to determine the severity of clinical conditions resulting from infection and for prognostic purposes. Animal models have been used to understand the mechanisms of the changes in homeostasis observed under pathological conditions. In the present study, we therefore report the changes in serum levels and hepatic gene expression of cytokines and chemokines in two different animal models of acute-phase responses. The acute-phase response is a transient emergency response aimed at preserving life and bringing about the changes necessary to reduce and repair tissue damage after the removal of damaging noxious agents. Our data suggest that the liver may be responsible for the increase in the serum levels of cytokines and chemokines as part of the body’s defense response to tissue damage. It is therefore doubtful that inhibiting this response at any stage after infection could improve the prognosis of patients. These results may help to interpret the laboratory changes observed in critically ill patients, as may be the case following SARS-CoV-2 infection.

**Abstract:**

A mild to moderate increase in acute-phase proteins (APPs) and a decrease in serum albumin levels are detected in hospitalized COVID-19 patients. A similar trend is also observed for acute-phase cytokines (APC), mainly IL6, besides chemokines (e.g., CXCL8 and CCL2). However, the source of the chemokines in these patients at different stages of disease remains to be elucidated. We investigated hepatic gene expression of CXC- and CC-chemokines in a model of a localized extrahepatic aseptic abscess and in a model of septicemia produced by the intramuscular injection of turpentine oil (TO) into each hindlimb or lipopolysaccharide (LPS) intraperitoneally (i.p.) in rats and mice (wild-type (WT) and IL6-KO). Together with a striking increase in the serum IL6 level, strong serum CXCL2 and CXCL8 concentrations were detected. Correspondingly, rapid (2 h) upregulation of CXCL1, CXCL2, CXCL5, and CXCL8 was observed in rat liver after intramuscular TO injection. The induction of the gene expression of CXCL1 and CXCL8 was the fastest and strongest. The hepatic CXC-chemokines behaved like positive APPs that depend on IL6 production by activated macrophages recruited to extrahepatic damaged tissue. Chemokine upregulation was greatly reduced in IL6-KO mice. However, IL6 was dispensable in the LPS–APR model, as massive induction of hepatic chemokines studied was measured in IL6-KO mice.

## 1. Introduction

The acute-phase response (APR) is the ancient response of the animal body to tissue damage caused by various noxious agents, such as trauma, burns, ischemia, and chemicals, but perhaps most commonly by pathogens, such as bacteria or viruses [1]. The purpose of this response is to concentrate all defense components to reduce blood loss by forming a provisional clot by means of platelets, fibrinogen, fibronectin, and vWF VIII to eliminate the damaged tissue, to initiate the repair process, and restore the lost tissue components. To this end, mediators released by the “attacked” area lead to the recruitment of cells produced in the bone marrow, mainly granulocytes, platelets, and macrophages, but also some lymphocytes [2,3,4]. Such mediators also induce a “systemic” response proportional to the extent and amount of damaged tissue. The components of such a response include, on the one hand, fever, dizziness, headache, lethargy, muscle and joint pain, loss of appetite, and sometimes loss of smell and taste [1,5], and a dramatic change in the synthesis of serum proteins in the liver (acute-phase proteins). This change is due to a well-organized switch in priorities from mass production of albumin to a massive increase in the production of several “major” acute-phase proteins, such as C-reactive protein (CRP), serum amyloid A (SAA) [6], hepcidin [7], lactoferrin [8], and lipocalin-2 [9], and of “minor” acute-phase proteins, such as fibrinogen, complement C3 and other members of the same system, ceruloplasmin, and alpha-1-antitrypsin, the major component of the second electrophoretic protein component [5]. This massive switch is enabled by the intracellular change in priority at the pre-translational level with the reduction of albumin gene transcription and the upregulation of acute-phase protein gene expression. The amino acids required for such a massive increase in protein synthesis are not supplied by food delivery to the digestive tract due to the temporary loss of appetite, as is usually the case, but by the catabolism of muscle proteins.

The COVID-19 pandemic has fueled discussion about the importance of elevated serum levels of acute-phase mediators (cytokines), such as IL6, TNF-alpha, and interferon-gamma (IFN-gamma), and chemokines, such as IL8, measured in patients with varying degrees of disease severity at the time of hospitalization [10,11,12,13], and whether the effect of these mediators can be reduced by the administration of specific antibodies [14]. In fact, it has been discussed in particular that the inhibition of some acute-phase processes could have negative consequences for viral elimination [15,16] and also delay tissue repair processes.

Chemokines are known to play a critical role in local inflammatory processes. They are chemotactic cytokines that are divided into four main classes based on their structural differences: C, CC, CXC, and CX3C. Chemokines are known to attract leukocyte populations [17]. Among them, CXC chemokines, such as CXCL1/KC, CXCl2/MIP2, CXCL5/LIX, and CXCL8/IL8 are known to attract neutrophils as a chemoattractant, whereas CXCL9, CXCL10/IP10, CXCL11, and CXCL12 mainly attract T lymphocytes, but also neutrophils, to some extent [18]. Accordingly, CC-chemokines are mainly chemoattractants for various subpopulations of leukocytes, e.g., monocytes, eosinophils, basophils, T lymphocytes, dendritic cells, natural killer (NK) cells, and, to a lesser extent, neutrophils. Among these, CCL2/MCP1 is known to be the major chemokine [17,19,20].

Previous studies have shown increased levels of chemokines followed by leukocyte recruitment [21,22] in the damaged liver or in the peritoneal cavity. However, a massive increase in hepatic gene expression and the serum levels of IL8 can be triggered by the induction of tissue damage at extrahepatic sites without recruiting granulocytes to the liver [23].

In the current study, we used two different models of acute-phase response. The intramuscular injection of TO is a sterile abscess model in which cytokines are released by the distant injured organ (muscle) into the bloodstream to induce APR in the uninjured liver both in rats and in mice to study eventual interspecies differences. LPS-induced APR is a systemic inflammation in which cytokines are released by the liver itself into the blood by activated resident macrophages [8]. The mouse LPS–APR model was chosen for two reasons: (a) rats are very resistant to LPS and (b) the availability of IL6-KO mice.

We show that the liver constitutively expresses chemokine genes that may be the target of acute-phase cytokines, such as IL6, when tissue damage occurs at extrahepatic sites. The activation of liver tissue macrophages may be responsible for the upregulation of hepatic chemokine gene expression independent of IL6.

## 2. Materials and Methods

Animals: Male Wistar rats of 8–12 weeks of age with a body weight of approximately 170–200 g were bought from Harlan–Winkelmann (Brochen, Germany). Adult male B6.129S2-Il6tm1Kopf (IL-6-KO) mice and wild-type adult male C57BL/6J mice (25–28 g) that were 8–12 weeks of age were purchased from Jackson Lab. (Bar Harbor, ME, USA). Both the rats and mice were held under standard conditions with a 12 h light–dark cycle and ad libitum access to fresh water and food. The animals were held according to the instructions of the institute, the German Animal Welfare Convention, and NIH guidelines.

Chemicals: The chemicals used in the present study were of analytical/molecular biology grade and purchased from commercial sources, as follows: the total RNA was extracted from the livers of mice after homogenization in Trizol (Invitrogen, Carlsbad, CA, USA). The primers were purchased from IBA Lifesciences (Goettingen, Germany). Quantitative real-time (qRT)-PCR primers, M-MLV reverse transcriptase, platinum Sybr green qPCR–UDG mix were obtained from Invitrogen (Darmstadt, Germany); and dNTPs, protector RNase inhibitor, Klenow enzyme, primer oligo dT15 for cDNA synthesis, and Salmon sperm DNA were obtained from Roche (Mannheim, Germany). All other reagents and chemicals were from Sigma–Aldrich or Merck (Darmstadt, Germany).

### 2.1. Induction of APR and Harvesting Blood and Liver Tissue

APR was induced, animals were sacrificed, and tissues were removed as described previously [24]. Briefly, the animals were anesthetized with ether. Subsequently, 500 µL of TO was administered intramuscularly into each hind limb, the right and left hind limbs of rats, and 100 µL to each hind limb of the mouse. In a separate experiment, mice were injected intraperitoneally with 50 µg of LPS (concentration of 2 mg/kg) dissolved in 100 µL of saline. Saline-treated rats and mice served as controls. The animals were then sacrificed under anesthesia, and the heparinized blood and livers were harvested. Blood was collected from the inferior vena cava of the treated and control animals, allowed to clot overnight at 4 °C, and then centrifuged at 2000× *g* for 20 min. The serum was then collected again and stored at −20 °C. Pieces of the livers were frozen in liquid nitrogen and stored at −80 °C until further use for protein and RNA isolation.

### 2.2. Enzyme-Linked Immunosorbent Assay (ELISA) for IL6 and Chemokines

For detecting the serum levels of chemokines (CXCL2 and CXCL8), the ELISA kits (Biosource, San Diego, CA USA) were used as instructed. The serum levels of IL6 were detected using Quantikines IL6 ELISA kit from R&D Systems (Wiesbaden, Germany). The samples were processed according to the manufacturer’s instructions.

### 2.3. RNA Isolation and Real-Time-PCR

Total RNA was isolated from liver tissue and cDNA was generated by the reverse transcription of 1 µg of total RNA, as described previously [20]. The sequences of the primers are provided in Table 1. The data were normalized with the housekeeping gene and the fold-change in expression was calculated using threshold cycle (Ct) values.

### 2.4. Protein Isolation and Western Blot Analysis

About 50 mg of frozen tissue was homogenized for protein isolation. Aliquots of the homogenates were stored at −20 °C until further use for Western blot analysis. Protein isolation and Western blot analysis were implemented as described previously [25]. Western blot was performed utilizing commercially available antibodies for CXCL1, CXCL8 (R & D systems, Wiesbaden, Germany), and Stat 3 (Cell Science, Cologne, Germany) utilizing ECL reagent. The quantification of p-STAT-3 was performed by densitometric analysis using ImageJ software.

### 2.5. Immunohistochemical Study

Immunohistochemical analysis was executed on 5 µm-thick cryosections of the liver as described [20]. Briefly, the cold acetone/methanol-fixed sections were washed with PBS. Afterward, the sections were treated with blocking solution (0.1% BSA and 10% FCS in PBS) for 1 h, and further incubated with the primary antibodies of MPO (Dako, Hamburg, Germany) and ED1 (Serotec, Duesseldorf, Germany) at 4 °C overnight. Non-immune serum was used as a negative control. The number of positive cells was counted from the liver sections.

Statistical analysis: The data were analyzed using Prism GraphPad 4 software (San Diego, CA, USA). Three to six animals were used for each experiment and all experimental errors were shown as the S.E.M. Statistical significance was calculated by one-way ANOVA and Dunnett post hoc tests. Significance was accepted at * *p* ≤ 0.05, ** *p* ≤ 0.01, and *** *p* ≤ 0.001.

## 3. Results

### 3.1. Measurement of Serum Cytokine and Chemokines Level

The serum levels of IL6 and chemokines (CXCL2 and CXCL8/IL8) were measured by ELISA from 1 to 48 h after the start of APR. The serum IL6 level reached a significant level at 4 h, with a maximum increase at 12 h (Figure 1, upper panel). The IL6 level dropped at 24 and 48 h, but remained significantly higher than the control value after TO injection.

Similarly, an increased chemokine level for CXCL8 was detected, with a maximum at 6 h with the onset of APR. The concentration of CXCL8 decreased, but remained significantly above the control value until 48 h. The serum CXCL2 levels followed the same pattern as that which was observed for CXCL8, but at a lower magnitude, with a maximum at 12 h and a subsequent decrease afterward. The magnitude of the increase in the serum CXCL8 levels in the TO-treated rats was higher than that of CXCL2 (Figure 1 lower panel). The liver is the source of almost all plasma proteins and previous results reported by Sheikh et al. [23] indicated that the expression of IL8 was higher in liver tissue than in muscle.

### 3.2. Study of Gene Expression of Chemokines in TO-Treated Rat Liver by RT-PCR

The gene expressions of CXCL1 (200-fold) and of CXCL8 (100-fold) showed early increases, which were significant at 2 h after the start of the experiment. The maximum induction of CXCL1 and CXCL8 was detected after 6–12 h. Thereafter, the mRNA expression of these genes significantly decreased. Moderate but significant induction of CXCL2 and CXCL5 was observed, with a maximum at 6 and 12 h, respectively, after intramuscular TO injection. The extent of the induction of CXCL1 and CXCL8 was much higher than that of CXCL2 and CXCL5. The CXCL10 expression increase at 1 h dropped from 2 to 36 h, with a significant decline at 12 h and an increase again at 48 h compared to the controls. CCL2 expression dropped from 1 to 4 h and showed a significant increase at 6 h, and remained slightly elevated from 12 h to 48 h compared to controls (Figure 2).

### 3.3. Changes in Gene Expression Levels of Chemokine in Wild-Type (WT) and IL6-Knock-Out (KO) in TO-Injected Mice

To further investigate the role of the major acute-phase cytokine (IL6), chemokine gene expression was analyzed by RT-PCR in the livers of WT and IL6-KO mice after intramuscular TO injection. Similar to the rat, a strong, time-dependent (up to 100-fold) induction of CXCL1 and CXCL8 was detected in the liver of the mice, which started 2 h after TO administration. The gene expression of both chemokines was still upregulated in WT mice until 24 h after TO administration. The massive increase in the gene expression of these two genes in the liver was almost completely abolished when IL6-KO mice were treated with turpentine oil (Figure 3). CXCL10 showed a significant increase in WT mice at 2, 4, and 24 h, while that in IL6-KO mice expression decreased at all time points after TO administration. A significant increase in CCL2 expression was observed in WT-mice at 2 and 4 h, but no significant changes were observed between WT and IL6-KO mice at later time points (Figure 3).

Interestingly, the constitutive expression of the CXCL1-gene was several orders of magnitude higher than that of the CXCL8 gene in the livers of both WT and IL6-KO mice, as judged by the Ct values (Appendix A). Indeed, detailed Ct value investigation is an important, indirect indicator to estimate the basal relative gene expression and detect the changes of gene expression in the liver of rats and mice under the different conditions of the acute phase [9,20].

Interestingly, the gene expression of another granulocyte chemoattractant, CXCL5 [26], the basal expression of which in mouse liver tissue is similar to that of CXCL8 (Appendix A), showed moderate, but significant, induction in liver tissue. A moderate increase in the gene expression of CXCL2 was also observed, with the maximum 12 h after TO injection in WT mice. The hepatic gene expression of all mentioned genes remained significantly lower in IL6-KO mice compared with WT mice (Figure 3).

CXCL10 and CCL2 showed no significant change in any of the animal groups after TO injection (Figure 3). In summary, the following results were obtained in this animal model of muscle tissue injury.

A.Chemokine-genes are constitutively expressed in liver tissue at different magnitudes;B.Changes in the expression of various chemokines in the liver can be divided into three groups:
(a)IL6-strongly upregulated (up to 150-fold) chemokines: CXCL1 and CXCL8 (independent of the constitutive amount of their mRNA);(b)IL6-moderately upregulated chemokines: CXCL5 and CXCL2;(c)IL6-unaffected chemokines: CXCL10 and CCL2.


### 3.4. Changes in Hepatic Gene Expression Level of Chemokines in WT and IL6-KO LPS-Injected Mice (Intraperitoneal)

In the second acute-phase response model, the septicemia-like model, an increase in hepatic CXCL1 and CXCL8 gene expression of a similar magnitude as that in the other model was measurable after intraperitoneal (i.p.) injection of LPS, but upregulation declined earlier. However, this decrease was significantly greater in the IL6-KO mice as early as 4 h after LPS injection. In contrast to the response observed after intramuscular TO injection, dramatic early induction of CXCL1, CXCL2, CXCL5, CXCL10, and CCL2 was observed, with a maximum 2 h after LPS administration. The gene expression of CXCL2 was highest (approximately 4000-fold), followed by CCL2 (3000-fold), CXCL10 (2500-fold), CXCL5, and CXCL1 in WT mice. The expression of all genes studied remained upregulated for up to 12 h and then decreased. The level of these chemokines decreased to almost normal in WT mice 24 h after LPS treatment. In contrast to TO-treated IL6-KO mice (muscular tissue damage), LPS injection resulted in an early and maximal increase (2 h) in all chemokine gene expressions, but a significant, drastic decrease in the mRNA levels of CXCL1, CXCL2, CXCL8, and CCL2 was observed in IL6-KO mice at 4 h and 6 h. Thereafter, the levels of these chemokines remained significantly lower compared with those of WT mice, but higher than those of control mice throughout the course of the study. The gene expression of CXCL5 remained upregulated until 12 h and then decreased sharply after 24 h compared with the control mice. RT-PCR showed no significant change in the gene expression of CXCL10 between the WT and KO groups (Figure 4).

In summary, the direct activation of liver macrophages by the intraperitoneal administration of a potent activator, such as bacterial lipopolysaccharide (LPS), resulted in a massive upregulation of the gene expression of chemokines that are not affected by IL6, as shown in the TO model of tissue injury (Figure 3). This suggests that other cytokines, such as IFN-gamma, IL1, and TNF-alpha, may play a major role. As shown by Malik et al. [20], myofibroblasts of hypoxia-“suffering” vessels may also become a source of acute-phase cytokines and chemokines, as seen in critically ill patients in the intensive care unit (ICU).

Of note, the constitutive gene expression of CXCL10 was comparable to that of CXCL1 in both mouse models (Appendix A).

### 3.5. Detection and Change in Chemokines and STAT-3-Protein Levels in Liver Tissue of WT and IL6-KO Mice after Intramuscular TO or Intraperitoneal LPS Administration

Liver is the source of most of the serum and plasma proteins under normal and APR conditions [1]. The protein levels (intracellular) of CXCL1/KC and/CXCL8/IL8 were examined in the livers of WT and IL6-KO mice. IL8 protein was detectable in the liver of the control animals, suggesting that this chemokine is constitutively expressed in the liver and that the liver contributes mainly to its serum level. The increase in IL8 gene expression in the liver after intramuscular TO administration was much higher than that in muscle, and an increase in IL8 concentration was also found in the supernatant of cultured hepatocytes stimulated with IL-6 [23].

We detected an increased amount of protein accumulated in the liver of WT and IL6-KO animals of both APR models. The quantitative extent of upregulation was not as impressive as that of the mRNA and serum protein levels because most of the intracellular protein is rapidly secreted into the blood. The protein levels began to increase at 2 h and remained elevated until 24 h after TO or LPS treatment, at least for IL8. The changes were somewhat less impressive for CXCL1, especially in the TO model and in the IL6-KO mice (Figure 5A,B).

Interestingly, the amounts of CXCL1 and CXCL8 proteins were exactly the opposite of those that would be expected based on the PCR-Ct values (Appendix A). This discrepancy may be due to the different strengths (specificity and concentration of immunoglobulins) of the antibodies used.

The protein concentration of STAT-3 in liver tissue was examined in both APR models.

No or very little phosphorylation of STAT3 was detected in the liver tissue of WT and IL-KO mice collected before the start of the experiments by Western blot analysis (Figure 5A,B). In the livers of WT mice, the p-STAT3 band was clearly visible as early as 2 h and remained significantly elevated until 6 h after TO or LPS treatment (Figure 5C,D). The band was more pronounced in LPS-treated mice than in TO-injected WT mice. In contrast, this band was only weakly present in the liver tissue of IL6-KO mice at 6 h and 12 h after TO injection, and 2 h and 4 h after LPS treatment (Figure 5A,B). However, the STAT-3 protein was constitutively expressed in the liver of WT and IL6-KO mice, so there was an increase in IL6-KO mice in both APR models after the start of the experiments.

### 3.6. Immunohistochemical Detection of Neutrophil Granulocytes in the Rat Liver after Intramuscular TO-Treatment

Indirect immunohistochemical labeling was used to detect MPO (neutrophil granulocyte marker) in normal rat livers and rats after TO treatment. Few NG+ cells were detected in the normal rat liver. The number of NG+ cells was not increased at any time point after TO administration (Figure 6A,B,E). When antibodies against ED1 (marker for macrophages) were used, ED1-positive cells were detected in the sinusoidal region of the normal liver. Similar to MPO+, the number of ED1+ cells did not increase after TO administration in rat livers (Figure 6C,D,F).

## 4. Discussion

In this manuscript, we report the changes in the serum levels and hepatic gene expression of cytokines and chemokines in two different animal models of acute-phase reaction. In the first model, tissue injury was induced by the intramuscular administration of turpentine oil, which caused a transient sterile abscess. Local tissue injury is characterized by the recruitment of polymorphonuclear cells, and also by the recruitment of mononuclear phagocytes. The release of cytokines into the systemic circulation caused lethargy, loss of appetite, and weight loss in the animals. The cytokines also induced dramatic changes in gene expression in the liver, characterized by the reduction of albumin synthesis [27] and by the upregulation of the gene expression of major and minor acute-phase proteins that play an important role in the first line of defense in various situations of acute tissue injury [5], such as muscle injury caused by turpentine oil administration or the intraperitoneal administration of lipopolysaccharide of gram-negative bacteria, in which proteins, such as complement components or lactoferrin, are important for clearance of circulating pathogens by the reticuloendothelial system [8].

In addition, fibrinogen and vWF are important components of other defense mechanisms, namely the formation of the provisional clot by the deposition of fibrin, fibronectin, vWF, and platelets at the site of tissue damage [28].

The downregulation of cytochrome p-450 components in the hepatocyte may be important for reducing the metabolic activation of toxic compounds.

The hepatocyte is also the source of several chemokines that were previously thought to play a mainly local role in the recruitment of inflammatory cells, the upregulation of which is controlled by major acute-phase cytokines, such as IL6 [23]. In this manuscript, we attempt to show that these cytokines and chemokines may be termed pro-inflammatory, but, in a positive sense, are thought to be the main drivers of the acute “emergency” defense response to clear the harmful noxious agents. It is also important to understand that the serum levels of IL6 and IL8 may be elevated, for example, in patients with cardiogenic shock, even if no infectious agents are present in the body [29].

This must be distinguished from the long-term persistent recruitment of inflammatory cells when the defense system was unable to eliminate the pathogens, as we know from viral infections of the liver by the hepatitis C virus and the hepatitis B virus [30,31]. In other disease patterns, such as Crohn’s disease or rheumatoid arthritis, the cause of the persistent massive recruitment of granulocytes into the intestinal mucosa or into the joints is still unknown, and therapeutic intervention can only be nonspecific via the inhibition of the local production of cytokines [32].

Using these two animal models, the sequence of acute changes in gene expression at the RNA and protein levels in liver tissue responsible for the serum levels of the corresponding proteins could follow closely after the start of the experiment. Similar experiences can be created with models of the induction of ischemic tissue damage. Tissue damage outside of the liver induces leukocytosis due to the strong upregulation of hepatic CXCL8/IL8 gene expression and the effect of serum IL8 on bone marrow. Thanks to the strong local and systemic acute-phase response, which quickly eliminates the causative noxious agent, recovery is usually completed within 96 h.

The TO–APR experiments performed in IL6-KO mice showed that the lack of IL6 led to a reduction in the hepatic gene expression of the chemokines studied. When LPS was administered, the gene expression of chemokines CXCL2, CXCL5, and CXCL10, which were less involved in the TO model, was dramatically upregulated early after the start of the experiment, and the absence of IL6 (IL6-KO mice) did not significantly alter this gene expression, at least at the early time points after LPS administration.

IF-gamma is one of the acute-phase cytokines that is upregulated earlier than TNF-alpha, IL6, and IL1, particularly when their production is activated directly in liver macrophages [33,34].

CXCL10/IP10 is a low-molecular-weight peptide (10 kDa) that is readily excreted in urine [35] and belongs to the CXC chemokine superfamily with pleiotropic immunological [36,37] and non-immunological [38] functions.

In humans, the measurement of CXCL10 has been shown to be clinically important as it correlates with the severity of various clinical conditions [31,39,40], including acute respiratory infections [41]. It is, therefore, not surprising that the determination of the serum CXCL10 levels has been performed in patients with COVID-19 [42,43]. Serum levels have been found to correlate with lung disease severity, but not with viral load [44,45]. In fact, viral load decreases shortly after the onset of symptoms and continuously thereafter, even in hospitalized patients who develop severe disease, including septicemia [46,47,48,49]. Although a high viral load at the time of hospitalization may predict the worst outcome, it is not clear how long viable virus is present (if present at all) in the lungs [50]. At the same time, laboratory findings characteristic of the acute-phase reaction may also be seen in patients who later develop hypoxia. Infection leads to systemic (multiorgan) damage after hospitalization, and pulmonary changes may become a cause of death, despite the absence of viral replication [51,52,53], as is the case in acute respiratory distress syndrome (ARDS) accompanied by generalized microthrombosis as a common histologic feature [54].

As CXCL10 is cleared in the urine [35], serum levels may be elevated by impaired renal function, which is common in critically ill ICU patients. Here, we show that CXCL10 is constitutively expressed in liver tissue with the same order of magnitude in wild-type and IL6-KO mice. The basal gene expression of CXCL10 in liver tissue was significantly higher than that of CXCL8. This suggests a possible significant inhibitory role of CXCL10 [12,54] in maintaining body homeostasis.

This suggests that the stimulus for the production of acute-phase mediators in serum may not only be the virus in the nose at the beginning of the infection, but also the lung and the other damaged (ischemia?) organs at the time of hospitalization, while the main source of acute-phase proteins (eventually including cytokines) is the liver. In fact, changes in the expression of the same genes can be observed in all organs [55] outside the liver, but only the liver (both macrophages and more importantly hepatocytes) has the capacity to increase the level of acute-phase proteins in systemic circulation.

In this manuscript, we also show that the massive upregulation of gene expression of several chemokines besides CXCL8 can be induced when liver mononuclear phagocytes are activated and the locally released cytokines can further activate the hepatocellular production of chemokines.

The phosphorylation of STAT-3, the protective transcription factor, was reduced in IL6-KO mice of both models but remained inducible. The activation of the STAT3 pathway was recently detected in the kidneys of COVID-19 patients, but viral or virus-like particles were not detectable [56], further underlining that cytokines and chemokines are released in absence of the triggering agent [57,58].

Elevated serum levels of IL6 and IL8 have been reported in various clinical conditions in children [59] and adults [60]. Very high concentrations have also been measured in the ascitic fluid of patients with ovarian cancer [61], where production can possibly be due to cancer cells, as we have previously shown [62].

The acute-phase response is, however, a temporary emergency response aimed at preserving life and inducing the changes necessary to reduce and repair tissue damage after the elimination of the damaging noxious agents. Interventions aimed at suppressing this response may even be dangerous. Indeed, we show that the synthesis of serum chemokines, as well as serum cytokines, is also upregulated in the liver (fortunately) independently of the serum levels of IL6. The data obtained for the IL6-KO mice suggested that inhibition of IL6 activity in response to tissue damage may not be indicated.

COVID-19 patients may clinically progress through different phases characterized by different symptoms [58]. In most cases, patients report fever and dry cough shortly after infection. The fever can be quite high (up to forty degrees Celsius) and last for varying lengths of time. At the same time, other symptoms, such as loss of smell and taste, may further reduce caloric intake.

Some of those who become ill are cared for in the emergency room, and some of them require oxygenation to the point of intubation. Many of them remain hospitalized, even after SARS-CoV-2 is no longer detectable. These patients may go through different phases of acute-phase reactions.

While the first mechanisms causing fever, joint pain, loss of appetite, and lethargy may last for a few days before hospitalization [49,50], other mechanisms may come into play later, especially during treatment in the ICU.

These different phases must be identified and thoroughly analyzed [51,52,53] to find the right therapeutic intervention [63]. In discussing the data obtained by measuring the serum cytokine and chemokine levels, it should be remembered that acute-phase response cytokines are important to reestablish homeostasis [64], albumin serum level measurement is important, and that the correction of hypoalbuminemia by albumin infusion is vital to treat hypoxemia [65].

Autopsy can help to understand the pathophysiology of this viral infection and eventually clarify the true extent of ischemic tissue [54], and also the source of cytokines and chemokines as the true cause of death [66]. This can only be achieved if clinicians and pathologists work closely together.

## 5. Conclusions

The increase in the serum chemokine levels (e.g., IL8) is most likely a time-dependent consequence of upregulated hepatic gene expression as a result of extrahepatic tissue damage. This increase is mediated by the major acute-phase cytokine, IL6. The activation of the intrahepatic acute-phase response by the intraperitoneal injection of lipopolysaccharide from the wall of Gram-negative bacteria is capable of eliciting a similar response, independent of IL-6.

These results may help to interpret the laboratory changes observed in critically ill patients after SARS-CoV-2 infection.

## Figures and Tables

**Figure 1 biology-11-00470-f001:**
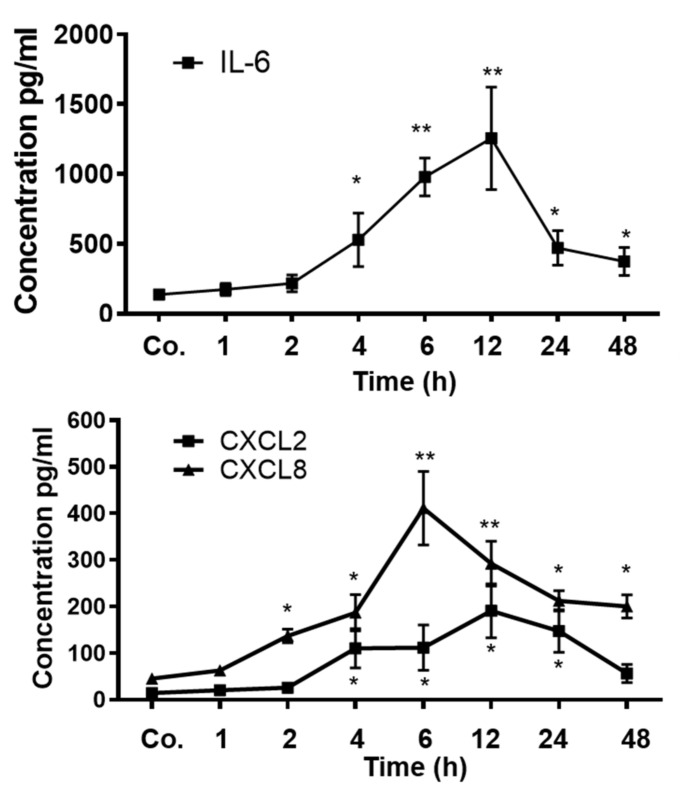
Measurement of IL6 and chemokines by ELSIA in rat serum after TO injection administered intramuscularly into the right and left hindlimb of rats at a concentration of 5 mL/kg. The data are compared with the controls (co), which received only saline into both hindlimbs. Both the treated and control animals were sacrificed at 1, 2, 4, 6, 12, 24, and 48 h after injection. Results represent the mean ± SEM values compared with saline-treated controls for each time point (analyzed by one-way ANOVA analysis of variance, *n* = 6). Significance was accepted at * *p* ≤ 0.05, ** *p* ≤ 0.01.

**Figure 2 biology-11-00470-f002:**
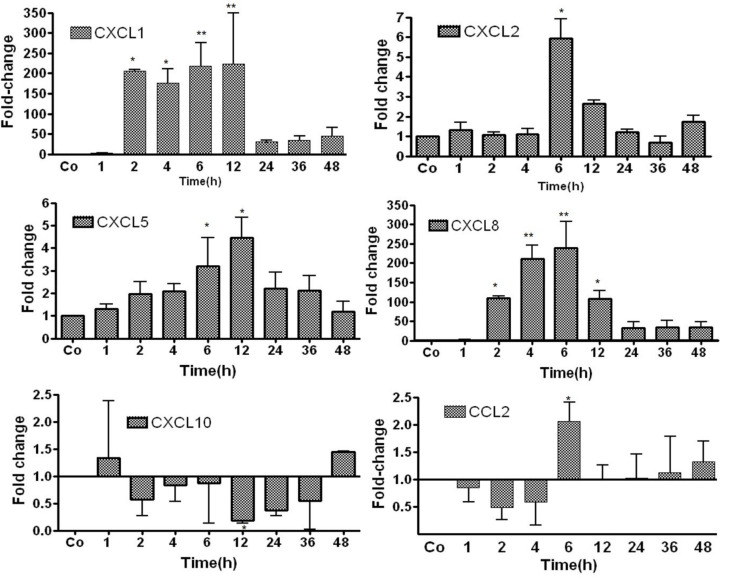
RT-PCR analysis of the total liver RNA from rats after TO injection administered intramuscularly into the right and left hindlimbs of rats at a concentration of 5 mL/kg. The data were compared with controls (co), which received only saline into both hindlimbs. Both the treated and control animals were sacrificed at 1, 2, 4, 6, 12, 24, 36, and 48 h after injections. The controls were set as 1. Fold-changes of mRNA expression of CXCL1 and CXCL2, CXCL5, CXCL8, CXCL10, and CCL2 are presented. Results represent the mean ± SEM values compared with saline-treated controls for each time point (analyzed by one-way ANOVA analysis of variance, *n* = 6). Significance was accepted at * *p* ≤ 0.05, ** *p* ≤ 0.01.

**Figure 3 biology-11-00470-f003:**
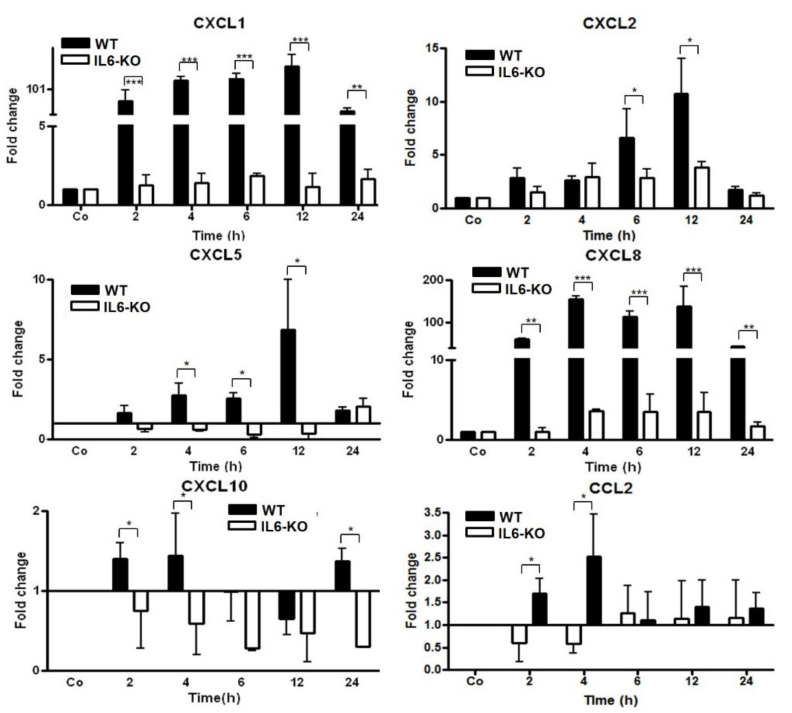
RT-PCR analysis of total liver RNA from WT- and IL6-KO mice after TO injection administered intramuscularly into the right and left hindlimbs of mice at a concentration of 100 µL on each hindlimb. The data were compared with controls (co), which received only saline into both hindlimbs. Both the treated and control animals were sacrificed at 2, 4, 6, 12, and 24 h after injection. The controls were set as 1. Fold-changes in the mRNA expression of CXCL1 and CXCL2, CXCL5, CXCL8, CXCL10, and CCL2 are presented. Results represent the mean ± SEM values compared with saline-treated controls for each time point (analyzed by one-way ANOVA analysis of variance, *n* = 6). Significance was accepted at * *p* ≤ 0.05, ** *p* ≤ 0.01, and *** *p* ≤ 0.001.

**Figure 4 biology-11-00470-f004:**
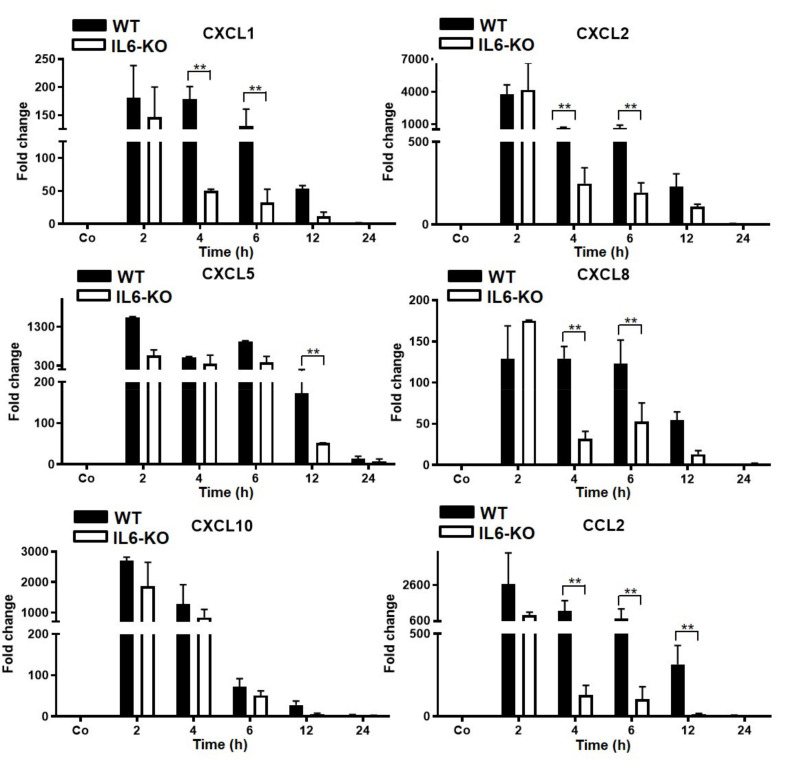
RT-PCR analysis of total liver RNA from WT- and IL6-KO mice after LPS injection intraperitoneally with a concentration of 2 mg/kg dissolved in 100 µL saline. Saline-treated animals (100 µL i.p.) served as controls (co). Both the treated and control animals were sacrificed at 2, 4, 6, 12, and 24 h after injection. The controls were set as 1. Fold-changes of mRNA expression of CXCL1 and CXCL2, CXCL5, CXCL8, CXCL10, and CCL2 are presented. Results represent the mean ± SEM values compared with the saline-treated controls for each time point (analyzed by one-way ANOVA analysis of variance, *n* = 6). Significance was accepted at ** *p* ≤ 0.01.

**Figure 5 biology-11-00470-f005:**
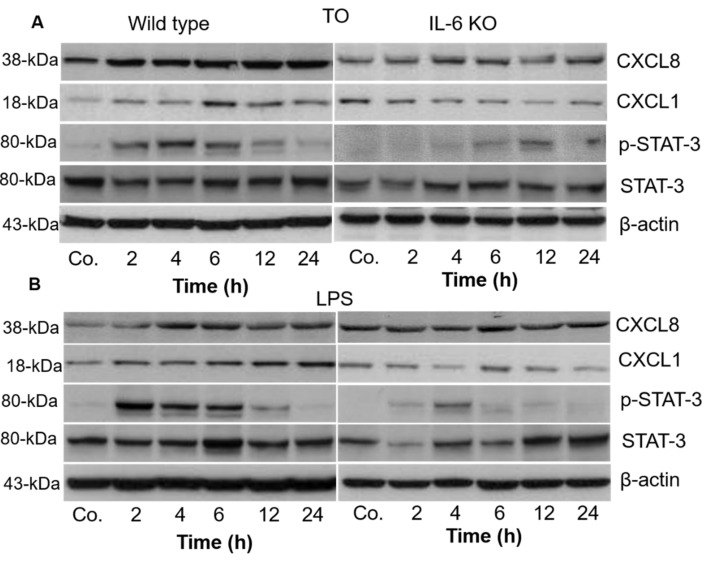
Western blot analyses of proteins from mouse livers with antibodies against chemokines and STAT-3 in wild-type and IL6-KO mice after TO (**A**) and LPS injection (**B**). β-actin served as a loading control. Quantification of p-STAT-3 was performed by densitometric analysis after TO (**C**) and LPS injection (**D**) in WT and IL6-KO mice. The data were normalized with the loading control from the same lane. Saline-treated animals served as controls (Co.). Both treated and control animals were sacrificed at 2, 4, 6, 12, and 24 h after injection (*n* = 3). Significance was accepted at * *p* ≤ 0.05, and ** *p* ≤ 0.01. Appendix A, The complete WB diagram is shown in the supplement materials.

**Figure 6 biology-11-00470-f006:**
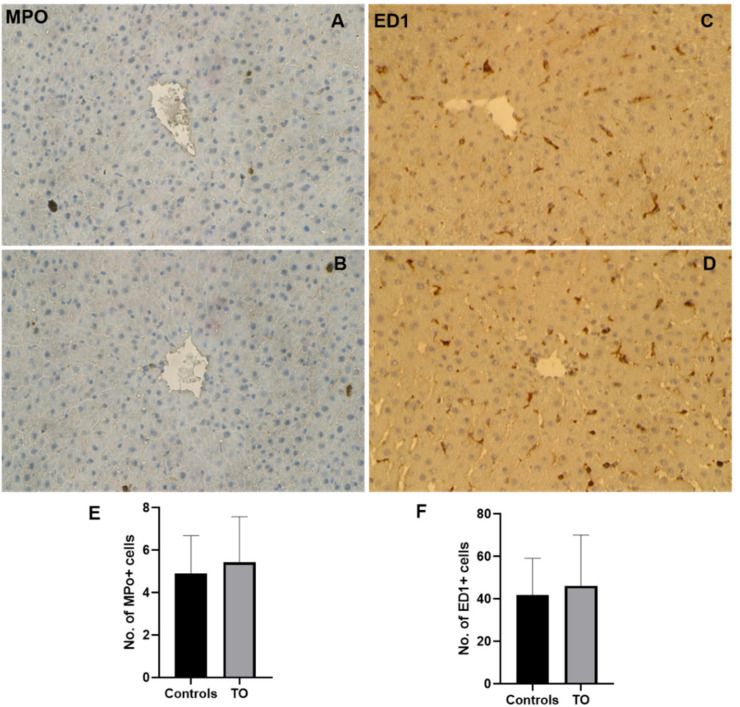
Immunohistochemical detection of MPO (marker for neutrophils) and ED1 (marker for macrophages) in rat livers after TO injection administered intramuscularly into the right and left hindlimbs of rats at a concentration of 5 mL/kg. Control livers, which received only saline (top) and TO-treated liver (bottom). Few MPO+ cells were detected in the control rat liver (**A**), which were not increased after TO treatment (**B**). ED1-positive cells were detected in the sinusoidal region of the control livers (**C**), and their number was also not increased after TO administration in rat liver (**D**). The presented image is representative of three experiments. Counted MPO (**E**) and ED-1 (**F**)-positive cells in the liver. Results represent the mean ± SEM values of three animals (*n* = 3) and six slides.

**Table 1 biology-11-00470-t001:** Primer sequences used in this study.

Primer	5 → 3 Forward	5 → 3 Reverse
(A) Rat primers		
CXCL1/Kc	GGCAGGGATTCACTTCAAGA	GCCATCGGTGCAATCTATCT
CXCL2/Mip2	ATCCAGAGCTTGACGGTGAC	AGGTACGATCCAGGCTTCCT
CXCL5/Lix	CTCAAGCTGCTCCTTTCTCG	GCGATCATTTTGGGGTTAAT
CXCL8/lL8	CCCCCATGGTTCAGAAGATTG	TTGTCAGAAGCCAGCGTTCAC
CXCL10/Ip10	CTGTCGTTCTCTGCCTCGTG	GGATCCCTTGAGTCCCACTCA
CCL2/Mcp1	AGGCAGATGCAGTTAATGCCC	ACACCTGCTGCTGGTGATTCTC
β-actin	ACCACCATGTACCCAGGCATT	CCACACAGAGTACTTGCGCTCA
Ubc	CACCAAGAAGGTCAAACAGGAA	AAGACACCTCCCCATCAAACC
(B) Mouse primers		
CXCL1/Kc	GGATTCACCTCAAGAACATCCAGAG	CACCCTTCTACTAGCACAGTGGTTG
CXCL2/Mip2	CTCTCAAGGGCGGTCAAAAAGTT	TCAGACAGCGAGGCACATCAGGTA
CXCL5/Lix	GGTCCACAGTGCCCTACG	GCGAGTGCATTCCGCTTA
CXCL8/IL8	GCTGGGATTCACCTCAAGAA	CTTTTGGACAATTTTCTGAACCA
CXCL10/Ip10	AAGTGCTGCCGTCATTTTCT	GTGGCAATGATCTCAACACG
CCl2/Mcp1	CCCACTCACCTGCTGCTACT	TCTGGACCCATTCCTTCTTG
β-actin	ATTGTTACCAACTGGGACGACATG	CGAAGTCTAGAGCAACATAGCACA
GAPDH	AGAACATCATCCCTGCATCC	CACATTGGGGGTAGGAACAC

## Data Availability

All data are included in the manuscript and Supplemental Data.

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
