# Peer review of "Interleukin-6-Production Is Responsible for Induction of Hepatic Synthesis of Several Chemokines as Acute-Phase Mediators in Two Animal Models: Possible Significance for Interpretation of Laboratory Changes in Severely Ill Patients"

_biology, 2022, doi:10.3390/biology11030470_

Round 1
Reviewer 1 Report
This is the second review of a paper that I rejected in the initial form in December 2021. Ihtzaz Malik and Giuliano Ramadori show that in response to LPS an TO administration CXC Chemokines (beside CXCL10) behave as positive APP in mice and rat. The authors increased the readability of the paper and included the lacking information on experimental details and statistics.
Author Response
We would like to thank the reviewer for the positive evaluation of our manuscript. We have revised the entire manuscript and improved the text and language as suggested.
Reviewer 2 Report
The manuscript titled “Interleukin-6-production is responsible for induction of hepatic synthesis of several chemokines as acute-phase mediators in two animal models: Significance for SARS-CoV-2-infection” describes CXC-chemokines expression in rats and mice (wild-type and IL6-KO) models treated with turpentine-oil or lipopolysaccharide to induce acute phase response. The authors further use ELISA, qPCR, and western blot to test the indicated genes or proteins expressed in the animal models. The IL6-KO mice might be the novelty in this study. However, the animal models that were used in the manuscript have no relationship with SARS-CoV2 patients, except the expression of some genes and proteins might have a similar pattern. Honestly, the manuscript was not well prepared. The followings are some concerns and the comments have been pointed out that the authors may want to consider. Major Concerns and Comments: 1) The animal models mentioned in this study, rats or mice were treated with TO or LPS, have no relationship with SARS-CoV2 patients. So the title needs to be reconsidered. 2) Line 17: The authors defined “APR” from “acute phase proteins” do you mean the letter “R” in “APR” from the word “proteins”? While in line 41, the authors define “APR” from “acute phase response”. 3) Line 91-93: “LPS induced APR is a systemic inflammation in which cytokines are released from the liver itself into the blood by activated resident macrophages.” I’d like to suggest the authors add related references here. Why did the authors choose TO to treat the animal model? I’d suggest the authors provide more information. 4) Line 119-120: Would you please provide some information or explain why each mouse received a double concentration of TO than a rat, rat 5 ml/kg vs mouse 10 ml/kg? Like the information you provide one 200 g rat was injected 1 ml TO, and a 20 g mouse was injected with 0.2 ml TO, that’s right? I’d suggest the authors provide the volume of reagent for each animal as well if applicable. 5) Line 121-122: “2 µg/mice” should be “2 µg/mouse”. Provide the volume of LPS you injected into each mouse, only 2 µg per mouse is not enough. 6) Line 123: Provide the procedure for how did you process the collected blood samples. 7) Line 188, Figure 1 legend: There is not a 36 h time point in the image, but you mentioned this time point in the legend. 8) Line 197-199: Results descriptions for CXCL10 and CCL2 were not matched with Figure 2. In the content, “The gene expression of CXCL10 and CCL2 showed no significant changes after TO-injection (Fig. 2). On the contrary expression of both genes seemed to decrease.” Figure 2 CXCL10 expression: increase at 1 h, dropped from 2 h to 36 h, come up again at 48 h, all compared with control. Figure 2 CCL2 expression: dropped from 1 h to 4 h, seems increased from 6 h to 48 h versus with control. Moreover, please double check and make sure the expression of CXCL10 at 12 h and 24h, CCL2 at 6 h versus control without significant difference. 9) Line 234-235: The description in the content does not match well with the CXCL10 and CCL2 expression shown in Figure 3. For example, the expression of CXCL10 in IL6-KO mice dropped at all time points. Please double check the statistics: CXCL10 at 24 h, CCL2 at 2 h and 4 h. Please provide detailed sample numbers. Only described as “three to six experiments” is not clear. 10) Line 282 Figure 4: CCL2 detection at 4 h and 6 h post treatment. In the wild type group, the values seem more stable with small error bars, while in the IL6-KO group the average values seem lower than 200-fold change but with huge error. Provide how many samples are in these two points and the value of each sample? The IL6 might affect the CCL2 mRNA expression in most mice. 11) Line 295-297: “IL8 protein was detectable in the liver of control animals, suggesting that this chemokine is constitutively expressed in the liver and that the liver contributes mainly to its serum level.” The authors concluded that the high level of IL8 in the serum is mainly from the liver. Please provide the evidence, references with descriptions to support this conclusion. 12) Line 309 Figure 5 western blot: Why some of the STAT3 with two bands some only with one band? Please provide necessary explanations and provide the original gel. 13) Figure 5 western blot: the authors compared the IL-6 KO with wild type on protein expression, while the internal control and all the targets proteins were separated into two different gels. All the β-actin bands in wild type group were overexposed than the IL-6 KO group. The same case for all other targets proteins. Based on this the analysis is not reliable. Repeat western blot on the same gel. Or please provide the original gel image if the authors originally run western blot on the same gel. Minor Concerns and Comments: 1) The text font sizes are not homogeneous throughout the manuscript. For example, the introduction part paragraph 1 and paragraph 2. 2) Line 105-106 Materials and Methods, Animals part: Provide mouse strain, gender, and animal age for rats and mice information. 3) Line 111-116 Materials and Methods, Chemicals part: Provide clear sources of chemicals in this study to make your experiments more reproducible by other researchers. 4) Line 159: I’d suggest the authors add “-“ between “one” and “way” as “one-way ANOVA”. 5) Line 160: The “P” should be lower case “p”. 6) Line 190: gene expression, line 191: gene-expression; 7) Line 191: “200-folds” should be “200-fold”, “100-folds” should be “100-fold”, and please check this throughout the whole manuscript. 8) Line 210 and throughout the whole manuscript: I’d suggest the authors use upper case “KO” for the knock-out instead of lower case, the same as you use WT short for wild type. 9) Line 223 and throughout the whole manuscript: I’d suggest the authors use lower case “t” in Ct-value as you used it in line 478. Capitalized “T” is needed for the first letter Table 1. 10) Line 213-220: No related Figure had been cited for the description of this result. 11) The authors use RT-PCR in line 214, while figure 2, figure 3, and figure 4 are qRT-PCR. 12) Line 244: “CXCL5” without “-“, “CXCL-2” with “-“, please homogeneous the style throughout the whole manuscript. 13) Line 249: “Reeated” in the sentence might be “treated”, “one-way” should be “one-way ANOVA”. 14) Line 295: “Il8” should be “IL8”. 15) Line 301: “2 hours” and “4 hours”, while in line 323 are like these “2-4 h” and “6-12 h”, and sometimes without space between the number and “h”, for example, see line 107 “12h”. Please check throughout the whole manuscript. 16) Line 313: “supp. table 2” the first letter “S” and “T” should be capitalized. 17) Line 320 and line 321: “(5AB)” should be “(Figure 5A and 5B)”, “(5CD)” should be “(Figure 5C and 5D)”, 18) Line 334: “ED-1”, while in line 335-336: “ED1”. 19) The authors did not mention what’s the meaning of “**” and “***” for the statistically significant in all the figure legends throughout the whole manuscript.Author Response
We would like to thank the reviewer for the very careful analysis of our manuscript and for the very constructive criticism.
- We agree that one of the main findings of our study is that the major APR cytokine, IL-6, is crucial for the dramatic increase in gene expression of several chemokines (e.g. IL8) in the liver. The latter is the second main finding of this study. This could place a primarily non-immunological organ at the center of the production of chemokines that become positive APPs, which is the third finding of the study.
1) Animal models should help to understand the mechanisms of the changes in homeostasis observed under pathological conditions. The increase in serum levels of IL-6 and IL-8 observed in patients primarily infected with SARS-CoV-2 virus has been used to determine the severity of clinical conditions resulting from infection and for prognostic purposes, suggesting that the virus itself is responsible for these changes (which mainly affect the lungs).
This study shows that the liver may be responsible for the increase in serum levels of cytokines and chemokines as part of the body's defence response to tissue damage. It is therefore doubtful that inhibiting this response at any time after infection could improve the prognosis of patients.
We have suggested a modified title: Interleukin-6-production is responsible for induction of hepatic synthesis of several chemokines as acute-phase mediators in two animal models: possible significance for interpretation of laboratory changes in SARS-CoV-2-infection.
2) Line 17. The reviewer is correct, thank you for that comment, indeed APR=Acute Phase Reaction and APP=Acute Phase Protein. We have now corrected the error.
3) Line 91-93. Thank you again, we have introduced the reference (Ahmad et al. 2011 Int J Biochem cell biol, Ref no. 8 in literature).
4) Line 119-120. Thank you again for this important question: while the rats were injected with 500 µl in each hind limb, the mice were injected with 100 µl for each hind limb, and it is true that there is no direct correlation with the difference in body weight between rats and mice (see lines 142-144).
5) Line 121-122. Thank you very much again the proper information has been introduced into the text (see lines 144-145).
6) Line 123. The information is now provided (see lines 146-149).
7) Line 188. The observation is correct, we have removed the 36 hours from the legend.
8) Line 197-199. Thank you for your careful observation. The new statistics and changes in the text have now been introduced as suggested (see lines 224-227 & Fig. 2).
9) Line 234-235. The new statistics and changes in the text have now been introduced as suggested (see lines 252-256 & Fig. 3).
10) Line 282. Fig .4 Thank again. The new statistics and changes in the text have now been introduced as suggested (see lines 301-303 & Fig. 4)
11) Line 295-297. The reviewer is right. The assumption arises from the general knowledge that the liver is the source of almost all plasma proteins and from previous results by Sheikh et al. (23) in which the expression of IL8 was higher in liver tissue than in muscle. The increase in IL8 gene expression in the liver after intramuscular TO administration was much higher than in muscle, and an increase in IL8 concentration in the supernatant of cultured hepatocytes stimulated with IL-6 was also detected. This comment has now been added to the text (See lines 209-211 & 334-337).
12) Line 309. According to the information provided by company STAT 3 has two isoforms, and the expression of each form is largely dependent on cell type, ligand exposure or cell maturation stage. Therefore, it is very possible to observe 2 bands for STAT-3. Please see the link below
https://www.cellsignal.com/products/primary-antibodies/phospho-stat3-tyr705-antibody/9131
13) Figure 5. The experiments with WT and KO mice were performed on different gels because we used pre-cast 10-well gels, which are standard gels used in laboratories to obtain clear and clean results. It was not possible to load all samples with marker into one gel. Therefore, we had to use different gels, however, both gels were run in parallel and treated in exactly the same way. However, normalized and quantified p-STAT-3 Western blot data using densitometer analyses has been shown (see Fig. 5CD).
Minor concerns:
1) Unfortunately, we could not see the difference in font size between paragraph 1 and 2. The formatting of the manuscript also corresponds to the journal's template, over which we have no control.
2) 105-106 The information has now been introduced.
3) The required information has now been provided.
4) Line 159. Corrected thank you for the kind suggestion.
5) Line 160. The correction has been made.
6) Line 190. “Gene-expression” corrected.
7) Line 191. 200-fold (correction made) thank you.
8) Line 210. IL6-KO has been corrected in the text.
9) Line 223. Ct is now in the text and Tables also.
10) Line 213-220. Figure 3 is now in the text (See line 252).
11) qRT-PCR has now been replaced with RT-PCR.
12) Line 244. The dash has been deleted.
13) Line 249. Treated, one-way ANOVA (Corrected).
14) Line 295. IL8 is now used throughout the text.
15) Line 301. Hours instead h is now used throughout the text.
16) Line 313. Corrected as suggested. Thank you!
17) Line 320 and line 321. Corrected as suggested.
18) Line334: Corrected “ED1”.
19) The information has now been provided in the legend of each figure.
Reviewer 3 Report
The Authors of the paper report the changes in serum levels and hepatic gene expression of cytokines and chemokines in animal models of acute phase reaction. They investigated hepatic gene expression of CXC and CC chemokines in a model of localized extrahepatic aseptic abscess and septicemia produced by intramuscular injection of turpentine-oil (TO) into each hindlimb or lipopolysaccharide (LPS) intraperitoneally (IP) in rats and mice (wild-type and IL6-ko) They show that the liver constitutively expresses chemokine genes that may be the target of acute-phase cytokines such as IL6 when tissue damage occurs at extrahepatic sites. Activation of liver tissue macrophages may be responsible for the upregulation of hepatic chemokine gene expression independent of IL6.
The relations of the obtained results to COVID-19 are interesting, but remain at the level of hypotheses. Text editing and language revisions should be made. I recommend the paper for publication after minor revision.
Author Response

(The authors gave the same response as above.)

Round 2
Reviewer 2 Report
With respect to the authors’ contributions, I carefully reviewed the revised manuscript several times and my comments list as below.
1: The circulating IL-6 and IL-8 are documented in patients with acute respiration diseases, cancer patients, heart disease, and so on. There is no surprise that the COVID-19 patients with high levels of IL-6 and IL-8.
The lists below are the articles about serum IL-6 and IL8:
A: Moreno-Guerrero SS, Ramírez-Pacheco A, Rocha-Ramírez LM, Hernández-Pliego G, Eguía-Aguilar P, Escobar-Sánchez MA, Reyes-López A, Juárez-Villegas LE, Sienra-Monge JJL. Association of Genetic Polymorphisms and Serum Levels of IL-6 and IL-8 with the Prognosis in Children with Neuroblastoma. Cancers (Basel). 2021 Jan 30;13(3):529. doi: 10.3390/cancers13030529. PMID: 33573284; PMCID: PMC7866803.
B: Lane D, Matte I, Rancourt C, Piché A. Prognostic significance of IL-6 and IL-8 ascites levels in ovarian cancer patients. BMC Cancer. 2011 May 30;11:210. doi: 10.1186/1471-2407-11-210. PMID: 21619709; PMCID: PMC3118896.
C: Hummel M, Czerlinski S, Friedel N, Liebenthal C, Hasper D, von Baehr R, Hetzer R, Volk HD. Interleukin-6 and interleukin-8 concentrations as predictors of outcome in ventricular assist device patients before heart transplantation. Crit Care Med. 1994 Mar;22(3):448-54. doi: 10.1097/00003246-199403000-00015. PMID: 8124996.
D: Lu YR, Rao YB, Mou YJ, Chen Y, Lou HF, Zhang Y, Zhang DX, Xie HY, Hu LW, Fang P. High concentrations of serum interleukin-6 and interleukin-8 in patients with bipolar disorder. Medicine (Baltimore). 2019 Feb;98(7):e14419. doi: 10.1097/MD.0000000000014419. PMID: 30762747; PMCID: PMC6407988.
2: After the first round review, the revised version improved the statistics part. Thank you for the authors taking my suggestions. However, there is no improvement or additional data provided to support the conclusion. For example the major concern 13 from the first review report about western blotting. I understand and do agree there might be some limited conditions to run all the samples in the same gel, however, this is not the excuse to try to find an additional method to solve the problem to obtain the major data to support the hypothesis of the research.
Author Response
1) We thank the reviewer for his contributions and for sending us the four very interesting publications.
We have now added the following sentence into the discussion section: It is also important to understand that serum levels of IL6 and IL8 may be elevated for example in patients with cardiogenic shock, even if no infectious agents are present in the body (Hummel et al. 1994) (See lines 418-420).
2) We thank the reviewer very much.
However, we believe that the picture is not crucial for the manuscript. The main message of the manuscript is that the liver is a potent and possibly the major source of serum chemokines such as IL8, MCP-1 and IP10 and that some of them are influenced by IL6 when tissue damage occurs at extrahepatic sites.
The upregulation of gene-expression in the liver can occur independently of IL6, as is the case when an acute phase situation is triggered in the liver itself by intraperitoneal injection of LPS (See lines 117 & 118).
We do not want to interpret criticism as an insinuation that the Western blot results were manipulated and that the experiment should be repeated for this reason. Moreover, in the latest versions we have provided densitometric analyses.
Round 3
Reviewer 2 Report
With respect to the authors’ contributions, my comments as below, of course, those are my personal opinions:
1: The articles I listed on the second-round review are documented evidence that circulating IL-6 and IL-8 are common in patients, there are so many related articles. There is no surprise that the SARS-CoV2 infected patients with high levels of them. In this case, the models that the authors provided had no relation with SARS-CoV2. The SARS-CoV2 in the title can be switched to cancer patients, patients with acute respiration diseases, patients with bipolar disorder, and so on. This is the top concern of the manuscript.
2: After the first-round and second-round reviews of my concerns about western blotting. With respect, and as the authors said “We do not want to interpret criticism as an insinuation that the western blotting results were manipulated”, I clarify that I have no meaning of that. The key point is we all come across the limited conditions of experiments; however, we have to try to find other ways to solve the problem. For example in the current manuscript, “TO model of CXCL8 and β-actin” bands, there were four times of exposure had been performed, one for wild type CXCL8, one for wild type β-actin, one for IL-6 KO CXCL8, one for IL-6 KO β-actin. The internal controls in the control group (wild type group) and treated group (IL-6 KO group) are completely separated into two different films, the same as target protein in wild type control and IL-6 KO groups are completely separated into two different films as well. This procedure completely increases the variation between the bands obtained, making the data incomparable.
Overall and with respect, to the current manuscript without solid data that directly support SARS-CoV2; there is no relation with the current pandemic. The manuscript is not of sufficient novelty.
Author Response
1. We thank the reviewer again for the kind suggestions. We have now modified the title and included the suggested publications into the manuscript (See lines 486-489).
2. Thank you very much for your comment. We understand the reviewer's concern and ideally, we would have done the same as suggested. As mentioned before, we used two gels for technical reasons, as we had more samples to load than could have been loaded on one gel. Therefore, we had to make one membrane from the WT group and a second one from the IL6-KO group. Both membranes were prepared in the same way and at the same time. Furthermore, the main aim of this experiment was to compare the untreated (control) mice with the treated mice of the WT or KO group, and indeed each membrane contained samples from untreated and treated mice in the WT or KO group. Moreover, we performed all blots on the same membrane (WT or KO) after stripping and using the target antibodies one after the other and quantified the p-STAT3 data by densitometric analysis to reduce any probability of deviations.
Although the focus of our manuscript was not to compare untreated WT and KO groups, we nevertheless presented CT values of PCR data to show that both WT and IL6-KO showed no difference in basal expression of the genes studied (See Table S1 and S2).
This manuscript is a resubmission of an earlier submission. The following is a list of the peer review reports and author responses from that submission.
Round 1
Reviewer 1 Report
In this paper, the authors measured protein and gene expression of IL-6, CXC and CC chemokines in serum and liver in rats and mice in tissue injury model using turpentine oil (TO) injection and septicemia model using lipopolysaccharide (LPS) intraperitoneal injection (IP). They reported increased serum IL-6, CXCL2 and CXCL8 and increased expression of CXCL1, CXCL2, CXCL5, and CXCL8, CXCL10 in rats in response to TO administration. Further, the authors found that these responses were decreased by using IL-6 KO mice. They also compared these findings with the LPS injection model. The methods and the results were well described.
The authors showed Immunohistochemistry data using MPO as a neutrophil marker and ED1 as a macrophage marker and stated that no differences were observed in the cellular accumulation between treatments. However, the data is not conclusive without the quantitative measurement of the signal. The authors also carried out the Western blot and stated that the pSTAT-3 is weak in IL-6 ko mice. However, it is not easy to make a concrete conclusion without a quantitative measurement of the bands. The authors also discussed the relevance of these findings with SARS COV2 infection. However, the authors did not discuss the possible differences in response to pulmonary infection compared to systemic responses.
Reviewer 2 Report
The manuscript "Interleukin-6 production is responsible for the induction of hepatic synthesis of several chemokines as acute phase mediators in two animal models. Significance for SARS-CoV2 infection" by Ihtzaz Malik and Giuliano Ramadori show that turpentine oil (TO) and LPS induce IL-6 secretion in rats and mice. TO-induced IL-6 in turn induced hepatic expression of chemokines (CC), suggesting that hepatic expression of these cytokines is regulated comparably to classical acute phase proteins. In contrast, LPS-induced IL-6 does not contribute to the early hepatic expression of chemokines (Fig. 4). These results are not innovative.
The title of this paper and the abstract suggest that the role of IL-6 and hepatic APP in SARS-CoV2 is being studied experimentally. However, this is not the case. The two mouse models used are classical models of aseptic inflammation and bacterial sepsis, respectively, and thus not a SARS-CoV2 model. There is nothing wrong with discussing the relevance of the results to the SARS-CoV2 pandemic, but especially in times of increasing scepticism towards scientists, such obvious deception for publicity reasons should not take place.
Moreover, the results section lacks a detailed description of the questions and the implementation of the results (see main points). The article lacks a clear description of the experimental data, lacks adequate controls and the conclusions are grossly over-interpreted, not only with regard to SARS-CoV2. The article is full of careless errors such as the inconsistent use of abbreviations and laboratory slogans.
In summary, I cannot recommend the publication of this article.
Key points:
Fig. 1: The exact scientific question and experimental procedure is not described in the text or in the caption. How much TO was injected? How was TO injected? Any information about the number of animals, the technical/biological replicates and the statistics performed is missing from the caption. It is unclear why some of the measurement points (IL-6 concentration) are connected with a dashed line. What does Co mean? Sham experiments controlling for the consequences of injecting TO are missing.
Fig. 2: The exact scientific question and experimental procedure is neither described in the text nor in the caption. How much TO was injected? How was TO injected? Any information about the number of animals, the technical/biological replicates and the statistics performed is missing from the caption. Some information in the text does not match the experimental data: For example, mRNA expression of CXCL2, CXCl5, CXCL8 is not significantly increased after 2h TO treatment. Sham experiments controlling for the consequences of injecting TO are missing.
Fig. 3. It is unclear to me why the authors switch from rats to mice. It would have made much more sense to already carry out the first experiments (Fig. 1 and 2) in mice. The reason for the detailed analysis of the CT values for the housekeeping gene and the genes of interest remains unclear. The presentation of the raw data in the form of CT values should be moved to the supplementary data and accompanied by a clear description of what is being presented. For example, it is unclear whether Table 2 (which should be called Table 1 as it is the first table) presents biological or technical replicates (mean ± SD?) and how many replicates were analysed. The title of the table is misleading as, for example, CXCL1 in IL-6 knockout mice was not analysed, but CXCL1 in IL-6 knockout mice.
Fig.4: The exact scientific question and the experimental procedure are neither described in the text nor in the caption. How much LPS was injected? Any information on the number of animals, technical/biological replicates and statistics performed is missing from the caption. The description of the Y-axis with odd numbers like 1351 is rather unconventional, unreasonable and somehow arbitrary for the different subfigures (cytokines). Again, the reason for the detailed analysis of the CT values for the housekeeping gene and the genes of interest remains unclear. The presentation of the raw data in the form of CT values should be moved to the supplementary data and accompanied by a clear description of what is being presented, e.g. it is unclear whether Table 3 (which should be called Table 2) presents biological or technical replicates (mean ± SD?) and how many replicates were analysed. The title of the table is misleading as, for example, CXCL1 in IL-6 co-mice was analysed rather than CXCL1-IL-6 double knockout mice. LPS mice is laboratory jargon.
Fig. 5. samples from wt mice and IL-6 ko mice must be analysed on the same gel. Otherwise (and as shown at the moment) changes in expression cannot be analysed at all.
Line 199: From the fact that IL-8 can be detected in the liver, it cannot be concluded that IL-8 is mainly produced in the liver.
Line 217: The authors claim that STAT3 phosphorylation was not analysed before the experiment (i.e. before injection of TO or LPS), but in Fig. 5 STAT3 phosphorylation is detected in the control animals?
References: Formatting of references is inconsistent, some (e.g. 1, 3, 8) are missing information on year of publication, journal, page numbers.
Spelling: authors should ensure consistent abbreviations throughout the text, e.g. IL-6 vs Il-6, Il-8 vs IL-8 vs Il8, Cxcl2 vs CXCL2 vs cxcl2,
Title: a dot is missing between models and significance.